# Combined segmentation and classification-based approach to automated analysis of biomedical signals obtained from calcium imaging

**Gizem Dursun[1], Dunja Bijelić[2], Neşe Ayşit[3], Burcu Kurt Vatandaşlar[3], Lidija Radenović[2], Abdulkerim Çapar[4], Bilal Ersen Kerman**[3,5,6], **Pavle R. Andjus[2], Andrej Korenić**[2]*, **Ufuk Özkaya**[1]

1 Electrical and Electronics Engineering Department, Süleyman Demirel University, Isparta, Turkey, 2 Center for Laser Microscopy, Faculty of Biology, University of Belgrade, Belgrade, Serbia, 3 Department of Medical Biology, Regenerative and Restorative Medicine Research Center (REMER), Research Institute for Health Sciences and Technologies (SABITA), School of Medicine, Istanbul Medipol University, Istanbul, Turkey, 4 Informatics Institute of İstanbul Technical University, İstanbul, Turkey, 5 Department of Histology and Embryology, School of Medicine, Istanbul Medipol University, Istanbul, Turkey, 6 Department of Medicine, Keck School of Medicine, University of Southern California, Los Angeles, CA, United States of America

* andrej.korenic@bio.bg.ac.rs

**Data Availability Statement:** All MATLAB files are available from the GitHub database, access via: https://github.com/gizemdursun/automated-

## Abstract

Automated screening systems in conjunction with machine learning-based methods are becoming an essential part of the healthcare systems for assisting in disease diagnosis. Moreover, manually annotating data and hand-crafting features for training purposes are impractical and time-consuming. We propose a segmentation and classification-based approach for assembling an automated screening system for the analysis of calcium imaging. The method was developed and verified using the effects of disease IgGs (from Amyotrophic Lateral Sclerosis patients) on calcium ($Ca^{2+}$) homeostasis. From 33 imaging videos we analyzed, 21 belonged to the disease and 12 to the control experimental groups. The method consists of three main steps: projection, segmentation, and classification. The entire $Ca^{2+}$ time-lapse image recordings (videos) were projected into a single image using different projection methods. Segmentation was performed by using a multi-level thresholding (MLT) step and the Regions of Interest (ROIs) that encompassed cell somas were detected. A mean value of the pixels within these boundaries was collected at each time point to obtain the $Ca^{2+}$ traces (time-series). Finally, a new matrix called feature image was generated from those traces and used for assessing the classification accuracy of various classifiers (control vs. disease). The mean value of the segmentation F-score for all the data was above 0.80 throughout the tested threshold levels for all projection methods, namely maximum intensity, standard deviation, and standard deviation with linear scaling projection. Although the classification accuracy reached up to 90.14%, interestingly, we observed that achieving better scores in segmentation results did not necessarily correspond to an increase in classification performance. Our method takes the advantage of the multi-level thresholding and of a classification procedure based on the feature images, thus it does not have to rely on hand-

analysis-of-biomedical-signals-obtained-from-calcium-imaging.

**Funding:** This work was funded by European Commission (EC) H2020 Marie Skłodowska-Curie Actions (MSCA) Research and Innovation Staff Exchanges (RISE) grant 778405 (Grant Recipient Prof. Pavle Andjus). The funders had no role in study design, data collection and analysis, decision to publish, or preparation of the manuscript.

**Competing interests:** The authors have declared that no competing interests exist.

crafted training parameters of each event. It thus provides a semi-autonomous tool for assessing segmentation parameters which allows for the best classification accuracy.

## 1 Introduction

Recently, automatic screening has become an essential part of the healthcare systems for assisting in disease diagnosing. In conjunction with machine learning-based methods, they became a popular topic in interdisciplinary studies. The categorization of medical imaging systems is based on two distinct steps: image reconstruction and image processing [1]. In the image reconstruction step, images are formed in two and/or three dimensions by using the projection data of an object. On the other hand, the image processing step involves enhancing image features (e.g., noise removal) and setting the features of images (e.g., segmentation) and their classification to perform object detection. Segmentation and classification are among the most popular techniques used in computer vision problems for object detection and interpretation for the majority of datasets [2, 3]. In particular, cell segmentation is the most popular topic for biomedical image analysis with the plethora of literature on segmenting different cell types [4–7]. Some of the most used methods in cell segmentation literature are thresholding, the active contour model, the watershed algorithm, and the deep learning approach [8–11]. The second most important topic for automatic screening systems is the classification for aiding in the organizing of biomedical image databases into categories [12]. A recent survey indicated that machine learning and deep learning algorithms, such as the Support Vector Machines (SVM) and the Convolutional Neural Networks (CNN), are the popular biomedical image classification methods [12].

Amyotrophic Lateral Sclerosis (ALS) is an adult-onset neurodegenerative disease that affects upper and lower motor neurons in the brain and spinal cord giving rise to both motor and extra-motor symptoms [13, 14]. The neuropathological hallmark of the disease is degeneration of motor neurons, infiltration of peripheral immune cells, and reactive gliosis surrounding degenerated neurons [15, 16]. In addition, as a sign of the activation of the systemic immune response, accumulation of immunoglobulin G (IgG) was observed in the spinal cord and cortical motor neurons of ALS patients [17]. Moreover, IgGs of ALS patients passively transferred to mice intraperitoneally, were taken up by the motor neurons causing an increase in the frequency of miniature endplate potential and the release of acetylcholine from the nerve terminal [18]. In the central nervous system, astrocytes enable neuronal survival by regulating neurotransmitter and ion homeostasis, energy metabolism, growth factor release, and blood-brain barrier formation [19]. However, astrocytes may also contribute by way of non-cell autonomous mechanisms to motor neuron damage in ALS [20–22]. Undoubtedly, intracellular calcium ($Ca^{2+}$) signaling plays several distinct roles in many physiological and pathophysiological processes in the nervous system in which astrocytes take part. Loss of the normal function of astrocytes elicits the disturbance of glutamate uptake by these cells and the motor neuron excitotoxicity by causing an accumulation of $Ca^{2+}$ in the synaptic cleft. Moreover, previously, we have shown that ALS IgGs caused an increase in the transient intracellular $Ca^{2+}$ levels thus observing an acute response in the cultured rat cortical astrocytes [23].

In the current study, we proposed a segmentation and classification-based approach for assembling an automated screening of cellular calcium signals. It was developed by using raw data of $Ca^{2+}$ fluorescence imaging in cultured primary rat cortical astrocytes upon treatment with IgGs isolated from patients diagnosed with the sporadic form of ALS (sALS).

Segmentation protocol was employed for detecting the boundaries of each astrocyte in the $Ca^{2+}$ imaging videos (i.e., time-lapse image stacks). Then, the traces were generated using pixels obtained at each time point within the detected boundaries. Subsequently, a classification step was applied to these traces (generated from all segmented astrocytes) to classify them as resulting from the treatment with either healthy or disease IgGs. Our method takes the advantage of the multi-level thresholding and of the classification procedure based on the feature images. These two approaches can be used complementarily to dynamically determine cells' boundaries, commonly called Regions of Interest (ROI), to allow for development of an easy bench to bedside automated screening.

## 2 Methods

### 2.1 Astrocyte primary cell culture and IgG isolation

Primary cortical astrocyte cultures were prepared from the cerebral cortices of neonatal rats (2-3 days old) and this procedure is described in more detail in Bijelić et al. [24, 25]. In brief, cells were grown in the culture medium (Dulbecco's Modified Eagle Medium based) and incubated at 37°C under a humidified 5% $CO_2$-containing atmosphere. Upon reaching confluence cells were subcultured and cultivated for up to 14 days prior to seeding onto coverslips for calcium imaging experiments. Cells were used in the experiments on the second and third day after the seeding. Animal procedures were approved by the Ministry of Agriculture, Forestry and Water Management Republic of Serbia, Veterinary Directorate, No. 323-07-11270/2020-05 and carried out in accordance with the strict protocols of the Ethics Committee for the Use of Laboratory Animals of the Faculty of Biology, University of Belgrade, Serbia (rsr. lic. 323-07-10457/2019-05), in the compliance with the National ethics committee–SLASA, as well as the EU Directive (2010/63/EU) on the protection of animals used for scientific purposes.

Blood samples were collected from patients clinically diagnosed with sALS (ALS group) and age-matched controls (non-ALS control group) at the Institute of Neurology, Clinical Center of Serbia and IgGs were purified at the Institute of Virology, Vaccines and Sera-Torlak, Belgrade, Serbia [23]. For the purpose of the AUTOIGG project (EC H2020 MSCA-RISE project No 778405) approval for the human subject research was obtained (850/6), as well as participant consent form which was implemented in the study.

### 2.2 Time-lapse fluorescence imaging

As previously described [25], cell-loaded coverslips bathed in extracellular solution (ECS containing in mM: 140 NaCl, 5 KCl, 2 $CaCl_2$, 2 $MgCl_2$, 10 D-glucose, and 10 HEPES; pH 7.4, 300 mOsm) were transferred into the recording chamber on the AxioObserver A1 microscope with an LD LCI Plan-Apochromat 25×/0.8 NA water immersion objective lens (Carl Zeiss). Intracellular calcium activity in astrocytes was assessed using Fluo-4 AM (Molecular Probes, Eugene, OR, USA). Astrocytes were loaded with 5 μM Fluo-4 AM up to an 1 h in ECS at room temperature. After rinsing, cells were kept in ECS for 15–30 min at room temperature to allow de-esterification of the dye. Fluo-4 was excited at 480 nm using Xenon Short Arc lamp (Ushio, Japan) coupled to the VisiChrome Polychromatic Illumination System (Visitron Systems GmbH, Puchheim, Germany). The emission light passed through the FITC filter set (Chroma Technology Inc., VT, USA) and it was recorded using an "Evolve" EMCCD 512 Digital Camera System (Photometrics, Tucson, AZ, USA), and VisiView® high-performance software (VisiChrome, Visitron Systems GmbH, Puchheim, Germany). During the experiments, cells were treated with human IgG fraction of peripheral blood collected from patients with sALS and non-ALS controls. Time-lapse images were acquired using a 1-Hz sampling rate for up to 15 min. Initially, fluorescence intensities were recorded for 1–3 min to determine the baseline

fluorescence. Thereafter, cells were treated for 5–10 mins by a bolus addition of IgG into the recording chamber to reach a final concentration of 100 μg/mL. Subsequently, IgGs were washed-out and astrocytes were observed after stimulation by ATP (1 mM for 5 sec) for up to 2 min. In this study, 33 videos were analyzed, of which 21 belonged to the disease and 12 to the control experimental groups.

## 2.3 The overview of the method

The main steps of the proposed method for assembling an automated screening system are shown in Fig 1. The method consists of three main steps: projection, segmentation, and classification. The code was written in MATLAB ver. 2022a (The MathWorks, Inc., Natick, Massachusetts, United States). First, the entire $Ca^{2+}$ time-lapse imaging recordings (videos) were projected into a single image using different projection methods. The final projected image represented the signal intensities of all pixels over time, and it was used for segmenting astrocytes. Next, the projected image was used in a multi-level thresholding (MLT) step and the boundaries of astrocyte ROIs were detected in this way. The ROIs were compared to the ground truths marked by the experienced researcher with expertise in calcium imaging on astrocytes cell culture. The performance of this step was evaluated using segmentation metrics. A mean intensity value of the pixels within these boundaries was collected at each time point thus producing the $Ca^{2+}$ traces (time-series). Finally, a new matrix called feature image was generated from those traces and used for classification. Different machine learning techniques including k-Nearest Neighbors (k-NN), Support Vector Machines (SVM), Decision Trees, and Ensembles were used in the classification step. Trained models were provided with the test data and the classification performances of different classifiers were compared.

**2.3.1 Projection.** Projection is usually known as mapping from a three-dimensional space to two dimensions. In our approach, the projection was used for mapping frames from $Ca^{2+}$ imaging videos onto a single image in two dimensions. Maximum intensity projection and standard deviation projection were the methods we implemented for the task of video projection. An example of applying different projection methods is presented in Fig 2. First, intensity values over the entire time axis for each pixel were determined. The maximum value was assigned to that pixel in the case of the maximum intensity projection image. In the standard deviation projection case, the standard deviation of intensity values over time for each pixel was calculated and it was assigned to that pixel in the projection image. Each frame of the video was a 16-bit image, thus consisting of 65,536 levels of grayscale with the range of pixel intensities of [0, 65535]. However, since the resulting range of pixel intensities in the projection image was rather narrow while an image created by the standard deviation projection method exhibited a very low contrast, a new image was created by extending the range of pixel intensities to the full range of [0, 65535] using linear scaling.

In summary, at the end of the projection step, all frames were mapped to the three different images by using maximum intensity projection, standard deviation projection, and standard deviation projection with linear scaling. All projection images were transformed to 8-bit images at the end of this step to decrease the computational cost of the next step.

**2.3.2 Segmentation.** The segmentation step we will describe here was applied to the images generated by the projection methods to obtain ROIs that could be assigned to individual astrocytes. Simultaneously, these areas were used to obtain $Ca^{2+}$ traces (time-series) for the subsequent classification step. First, the noise of each image was reduced by using a mean filter size of 5×5, resulting in more homogeneous regions. Multi-level thresholding (MLT) was implemented to determine the boundaries of each astrocyte. It is the basic method for image segmentation, and it is conducted by applying different levels of threshold to the projection

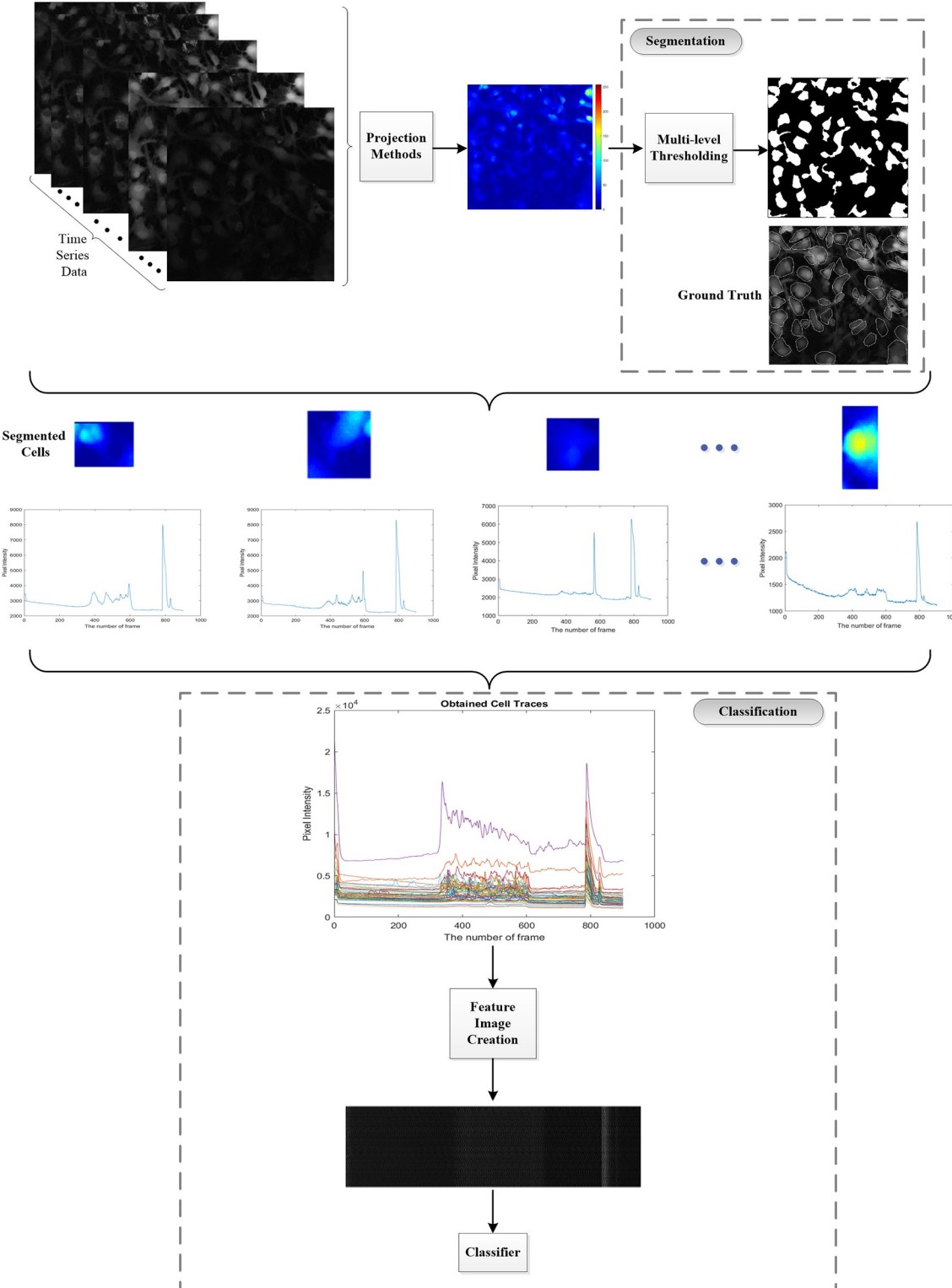

**Fig 1. Procedure steps of the proposed approach.** The entire Ca$^{2+}$ time-lapse imaging recordings (videos) were projected into a single image using different projection methods. Segmentation was performed by using a multi-level thresholding, and obtained binary mask was compared to the ground truths (ROIs marked by researcher). Finally, a new matrix called feature image was generated from extracted traces and used for assessing classification accuracy of various classifiers.

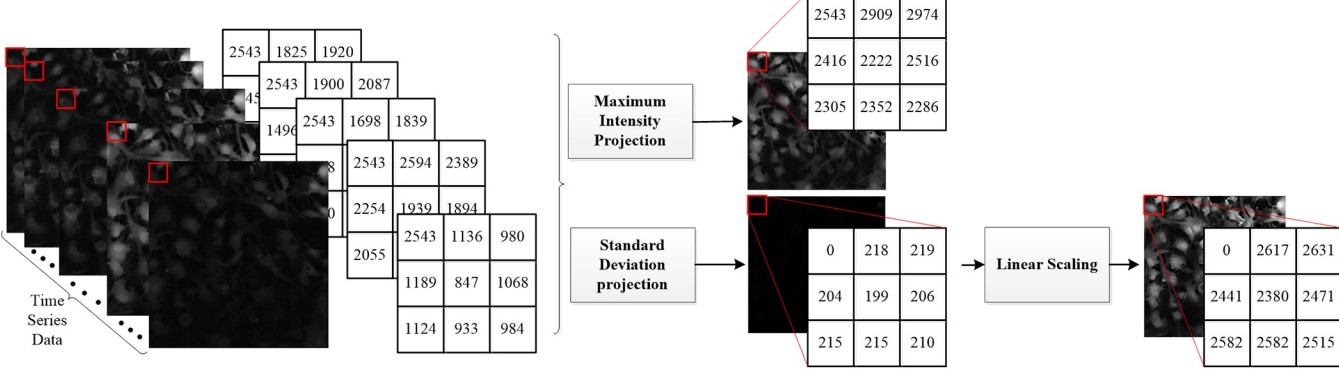

**Fig 2. The resulting images after applying different projection methods to intensity values over the entire time axis for each pixel of every frame.** 3×3 square matrices demonstrate the intensity values of original frames (left side) and resulting images (right side).

image. A threshold level was applied starting from the highest intensity value of 255 and it was successively decreased by 1 at each further step. In this manner, a binary image was obtained for every thresholding step and cumulative changes in that binary image were monitored for the appearance of new regions in the image (by "region" we assume pixels valued as 1) that were further defined as candidates for a cell. Namely, MLT focuses on determining a starting point (for each ROI) around which the region would subsequently increase its size. Such area was further monitored for changes with the application of subsequent thresholds.

The growth of all candidate cell regions along with the change of the threshold level was monitored, and these regions were allowed to enlarge according to the optimized criteria. These criteria were devised in accordance with the properties of ground truth ROIs, for example, one of them being the mean area of a cell. First and foremost, we aimed to obtain segmentation results that are as close as possible to what the researcher had marked. An example of implementing the multi-level thresholding method is shown in Fig 3. As a result, there were two scenarios for enlarging the area of candidate cell regions. In the first one, the candidate region can remain to be a monolithic area across all threshold levels, and thus it was allowed to expand until the area of that region has reached a pre-defined stopping criterion (see example of the progression of two cells in Fig 3B–3E). In the second one, the candidate cell region can merge with another expanding candidate cell region (Fig 3B–3D, top left part). In this scenario, areas of both regions were calculated, and they were not merged if the total area was

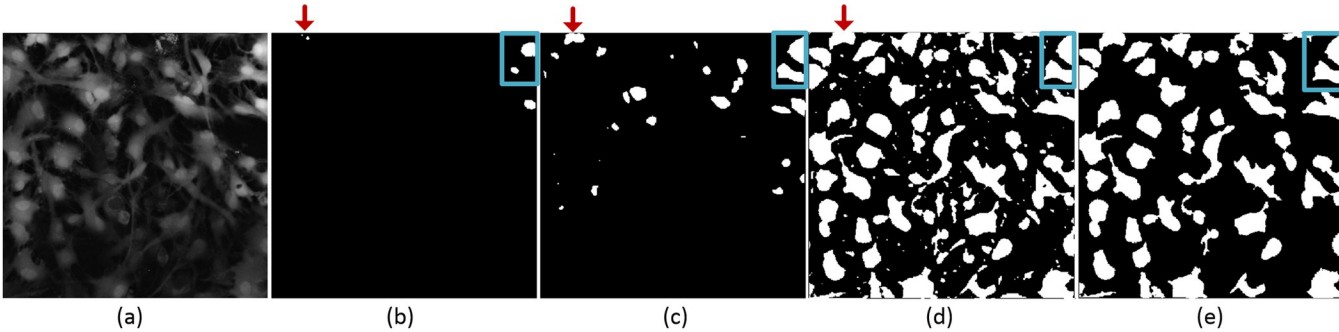

| (a) | (b) | (c) | (d) | (e) |

**Fig 3. An example of implementation of multi-level thresholding.** *a)* the maximum intensity projected image, *b-d)* three binary images obtained from three different threshold levels, *e)* the result of the multi-level thresholding method with elimination. The candidate cell region can merge with another expanding candidate cell region (top right corner of *c* and *d*).

larger than the selected junction criterion. In the opposite case, these two candidate cell regions were merged, and the merged region was allowed to expand until its area reached the pre-defined stopping criterion.

All these processes were applied to the projection image until the threshold level decreased from 255 to 0. At the end of this phase, our procedure identified some candidate cell regions that did not expand sufficiently to be considered as real cell regions, or they did not merge with other regions. In other words, applying multi-level thresholding also results in some noisy regions at the end of this phase (Fig 3D). To solve this problem, an elimination criterion was devised. If the area of the candidate cell region was smaller than the elimination criterion, this candidate region was discarded (Fig 3E). Thus, all regions, observed as a set of pixels, were detected using multi-level thresholding and the corresponding boundaries of each astrocyte were therefore detected automatically.

At the end of this stage, to assess the performance metrics of segmentation, precision and recall were calculated using these boundaries and ground truths. The goal was to evaluate at what extent the regions detected by MLT, were compatible with real boundaries of the astrocytes. Namely, one way to determine the quality of the resulting ROIs was to compare them to the ROIs previously identified by the researcher. In this regard, the optimization was aimed at the highest F-score. Precision, recall, and F-score were calculated in the following manner:

$$Precision = \frac{TP}{TP + FP} \tag{1}$$

$$Recall = \frac{TP}{TP + FN} \tag{2}$$

$$F - score = 2 \times \frac{(Precision \times Recall)}{(Precision + Recall)} \tag{3}$$

$$IoM = \frac{area(S \cap GT)}{\min(area(S), area(GT))} \tag{4}$$

where TP, FP, FN, S, and GT correspond to the True Positive, False Positive, False Negative, Segmented region, and Ground Truth, respectively. All metrics were calculated using a different threshold applied to the Intersection over Minimum (IoM) calculated between segmented regions (S) and ground truths (GT). If the calculated IoM value between the S and GT was larger than a defined threshold level, this region was counted as a true positive (TP).

**2.3.3 Classification.** In order to find the most efficient classifier of the $Ca^{2+}$ traces obtained from ROIs, we devised a method depicted with a flowchart in Fig 4 –a feature image was created from a segmented image obtained after applying multi-level thresholding. The intensities of all pixels within the detected ROI (obtained during the segmentation phase) were used to calculate the mean value of these pixels. This process was repeated for all frames of the

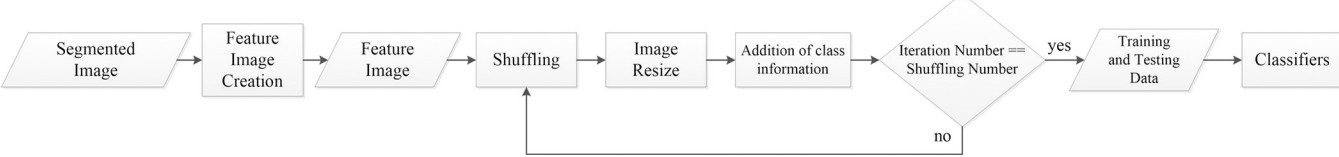

**Fig 4. Flowchart of the classification phase of our method.** It is emphasizing data augmentation steps necessary to combine with the segmentation into an automated screening system.

$Ca^{2+}$ imaging videos and the average pixel intensity obtained at each time point was stored as an element of an array vector. In other words, a row vector was generated for each detected ROI and each value of this vector represented the average pixel intensity value within the detected ROI at a particular time point. In this way, the matrix of a feature image was formed by storing the row vector for each detected ROI in successive rows. Therefore, a feature image has as many rows as the number of detected ROIs and as many columns as time points. For this reason, the size of the feature images differs from video to video according to the number of detected ROIs and the number of time points.

Shuffling was implemented to increase the number of data that were used for classification and image resizing was implemented to standardize the size of all feature images. Since during a single recording (i.e., experiment) astrocytes were treated with only one sample (control vs. disease), all ROIs in the $Ca^{2+}$ imaging video were belonging to the same class. In the shuffling step, the order of rows of the feature image was randomly mixed, thus eliminating the dependence between the ROIs, as if the resulting image was obtained from another experiment belonging to the same class. The number of times the shuffling procedure was applied was set to 200. If the feature size is variable as in our study, then it is acceptable to select the mean length of the feature size as the normalized length. The normalized height of the feature image was the average number of ROIs per video, which was around 32. Therefore, a shuffled feature image was downsized to 128×32 (with respect to the average width × height ratio) by using bicubic interpolation and then flattened to a vector (of size 1×4096), serving as the training and testing dataset. All the training and testing data were generated by a procedure that included shuffling, image resizing and flattening. This data augmentation process is important both to make training invariant to the order of ROIs and to increase the number of images required to train a supervised classifier.

At the end of the data augmentation process, we had generated 6600 shuffled and resized feature images. The images were then randomly divided into the training data (66.7%) and the test data (33.3%). Finally, a label, zero for healthy or one for the disease, was added as the last element of the shuffled and resized feature image vector. Table 1 shows all classifiers used in this study. Classifiers were trained using the training data and a trained model was evaluated against the test data. The accuracy of the testing step was calculated by using the added information regarding the class. At the end of the classification step, the calculated accuracies of the classifiers were compared, and the most efficient classifier was determined as the one with the highest score. We also checked the effect of the criteria governing the growth of candidate ROIs on classification accuracy. In other words, in this regard, the optimization criteria were to maximize the classification accuracy.

## 3 Results

The procedure steps shown in Fig 1 were implemented for segmenting the boundaries of astrocytes in $Ca^{2+}$ imaging videos and the classification of the obtained traces. Out of 33 videos we

**Table 1. List of classification methods.**

| Support Vector Machines | k- Nearest Neighbors | Ensemble | Decision Trees |
|---|---|---|---|
| Linear SVM | Fine k-NN | Boasted Trees | Fine Tree |
| Quadratic SVM | Medium k-NN | Bagged Trees | Medium Tree |
| Cubic SVM | Coarse k-NN | Subspace Discriminant | Coarse Tree |
| Fine Gaussian SVM | Cosine k-NN | Subspace k-NN | |
| Medium Gaussian SVM | Cubic k-NN | RUSBoosted Trees | |
| Coarse Gaussian SVM | Weighted k-NN | | |

**Table 2. Comparison of average F-score results for MLT for the range of junction criterion values given different IoM threshold levels (no other criteria applied).**
F-scores were averaged over all three projection methods (maxint, std, stdscale).

| IoM threshold | Average F-Score for Multi-Level Thresholding | | | | | | |
|---|---|---|---|---|---|---|---|
| | Junction Criteria Threshold | | | | | | |
| | 100 | 300 | 500 | 750 | 1000 | 1250 | 1500 |
| **0.1** | 0.7366 | 0.8567 | 0.8583 | 0.8218 | 0.7669 | 0.6979 | 0.6416 |
| **0.2** | 0.6363 | 0.7854 | 0.8159 | 0.7984 | 0.7480 | 0.6807 | 0.6206 |
| **0.3** | 0.5142 | 0.6635 | 0.7150 | 0.7086 | 0.6664 | 0.6035 | 0.5472 |
| **0.4** | 0.3956 | 0.5196 | 0.5576 | 0.5567 | 0.5169 | 0.4600 | 0.4126 |
| **0.5** | 0.2706 | 0.3572 | 0.3838 | 0.3860 | 0.3587 | 0.3152 | 0.2795 |

analyzed, 21 belonged to the disease and 12 to the control experimental groups. First, all videos were transformed into three projection images using the methods of maximum projection, standard deviation, and standard deviation with linear scaling. Second, multi-level thresholding was applied to each of these three projection images to detect the ROIs that would correspond to the boundaries of astrocytes.

In our study, we applied six different IoM threshold levels (0.1, 0.2, 0.3, 0.4, 0.5, and 0.7) and the segmentation metrics were calculated for each level separately. As presented in Table 2, for each IoM threshold level, the best segmentation results were obtained upon reaching the value of 500 for the junction criterion, and its further increase tended to decrease the segmentation performance. After we had determined the value for the junction criterion, we introduced the other two above-mentioned criteria, stopping and elimination, which were determined to be 3000 and 500, respectively.

All metric results for 33 $Ca^{2+}$ imaging videos are graphed in Fig 5. Metric results take values between 0 and 1, and the performance of segmentation increases as that value gets closer to one. The average F-score for all the data was about 0.86 throughout the first 5 threshold levels. For the last threshold level (IoM = 0.7), the average F-score dropped to 0.80 for all videos for all projection methods. Junction, stopping, and elimination criteria were set to 500, 3000 and 500, respectively. The F-scores are also presented in **Table 3** to compare the performances of different projection methods when only junction criterion was applied. Considering these results, the F-score using the maximum intensity projection was the closest to 1. Moreover, applying linear scaling to the standard deviation projection method increased the resulting F-score values.

After applying the segmentation step for detecting ROIs that correspond to the boundaries of astrocytes, feature images were created using extracted traces from these ROIs. Furthermore, shuffling and resizing were applied to feature images, and data for the classification step were generated. In this step, a total of 21 feature images belonged to the disease group, of which 15 were used for training. Similarly, 7 of the 12 feature images that belonged to the control group were used for the training. Classifiers listed in Table 1 were trained using the training data and the trained model was evaluated against the test data. Accuracies of selected classifiers were calculated and compared to find the most efficient classifier for the proposed method. For this purpose, 5-fold cross-validation was used for the evaluation of the performance of trained classifier models. Train and test accuracies for all classifiers are presented in Table 4.

Coarse Gaussian SVM, applied to training data generated by using the standard deviation projection method, was the most efficient classifier based on the test accuracy evaluation. Although the classification accuracy reached up to 90.14% for this projection method (Table 4), interestingly, the other two methods achieved better segmentation performances

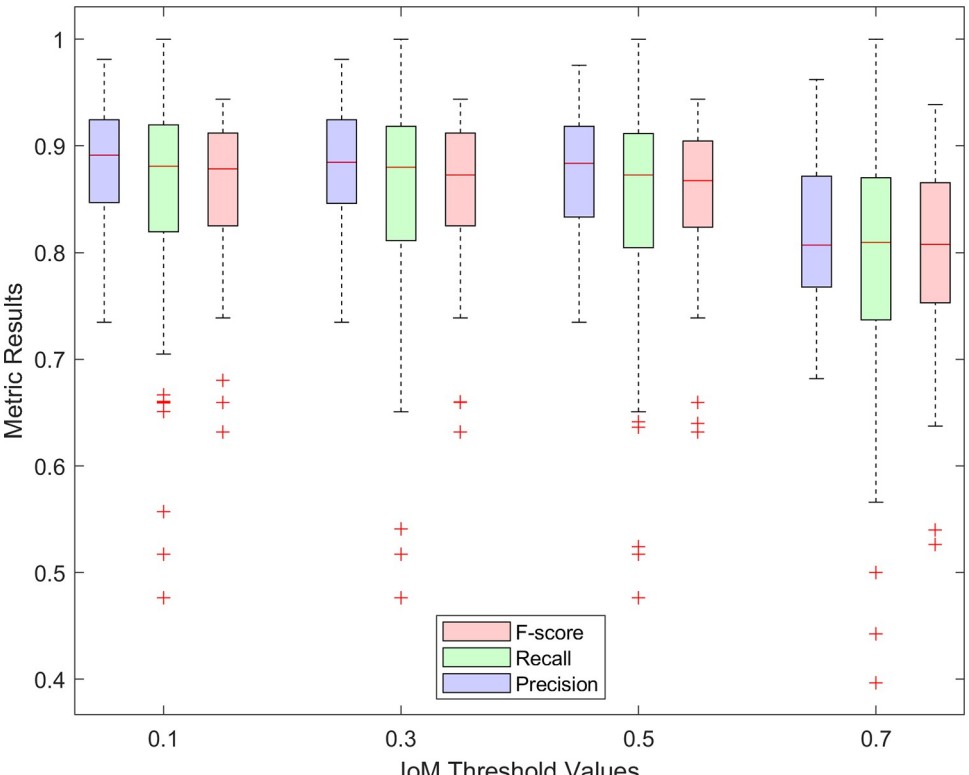

**Fig 5. Box plots of the statistical values corresponding to segmentation performance metrics: Precision, recall, and F-score.** Since all metric results take values between 0 and 1, relatively high and stable values of Recall and Precision for IoM values up to 0.5 justify the further usage of that level. Junction, stopping, and elimination criteria were set to 500, 3000 and 500, respectively.

(Table 3). At the same time, the most successful classification accuracy with training data created with the standard deviation with linear scaling projection method, was also obtained with SVMs. On the other hand, the Decision Tree served a successful classifier for the use of the maximum intensity projection method with an accuracy reaching up to 86.0%.

## 4 Discussion and conclusion

In this paper, a combined segmentation and classification-based approach is proposed for establishing an automatic screening system for $Ca^{2+}$ dynamics in an astrocyte cell culture. To that end, several excellent review papers have already covered the comparison of various methods for the analysis of astrocyte intracellular $Ca^{2+}$ fluctuations that can occur either under experimental stimulation or spontaneously [26–30]. However, our approach markedly differs in a way that, as a starting point, a segmentation method was coupled with the classification

**Table 3. Comparison of F-score results for different projection methods depending on the IoM threshold level (only junction criterion was applied).**

| Projection Method | IoM Threshold Level | | | | | |
|---|---|---|---|---|---|---|
| | **0.1** | **0.2** | **0.3** | **0.4** | **0.5** | **0.7** |
| **Maximum Intensity (maxint)** | 0.8745 | 0.8424 | 0.7563 | 0.5957 | 0.4352 | 0.1175 |
| **Standard Deviation (std)** | 0.8370 | 0.7811 | 0.6491 | 0.4826 | 0.2956 | 0 |
| **Standard Deviation with linear scaling (stdscale)** | 0.8636 | 0.8244 | 0.7397 | 0.5948 | 0.4206 | 0.1102 |

**Table 4. Train and test accuracy for selected classifiers for different projection methods.**

| Classifiers / Projection Methods | | Maximum Intensity | | Standard Deviation | | Standard Deviation with linear scaling | |
|---|---|---|---|---|---|---|---|
| | | Training Acc. (%) | Test Acc. (%) | Training Acc. (%) | Test Acc. (%) | Training Acc. (%) | Test Acc. (%) |
| Support Vector Machines | Linear SVM | 100 | 72.73 | 100 | 72.73 | 100 | 72.73 |
| | Quadratic SVM | 100 | 81.82 | 100 | 81.82 | 100 | 81.82 |
| | Cubic SVM | 100 | 70.64 | 100 | 63.64 | 100 | 72.73 |
| | Fine Gaussian SVM | 68.2 | 54.55 | 68.2 | 54.55 | 68.2 | 54.55 |
| | Medium Gaussian SVM | 100 | 81.82 | 100 | 89.59 | 100 | **81.82** |
| | Coarse Gaussian SVM | 95.0 | 80.77 | 86.6 | **90.14** | 90.9 | **81.82** |
| k- Nearest Neighbors | Fine k-NN | 85.5 | 60.27 | 86.9 | 64.27 | 82.3 | 65.23 |
| | Medium k-NN | 79.6 | 58.91 | 75.7 | 63.95 | 75.5 | 64.41 |
| | Coarse k-NN | 72.1 | 54.91 | 70.2 | 67.45 | 73.0 | 54.86 |
| | Cosine k-NN | 88.6 | 75.68 | 92.2 | 72.82 | 88.2 | 81.64 |
| | Cubic k-NN | 83.5 | 68.68 | 82.1 | 59.95 | 81.8 | 68.32 |
| | Weighted k-NN | 82.5 | 59.68 | 79.0 | 64.73 | 77.5 | 63.91 |
| Ensemble | Boasted Trees | 87.2 | 69.77 | 86.5 | 72.09 | 85.3 | 69.59 |
| | Bagged Trees | 82.7 | 67.45 | 81.2 | 71.09 | 80.9 | 68.91 |
| | Subspace Discriminant | 75.9 | 58.5 | 78.0 | 69.05 | 74.9 | 62.82 |
| | Subspace k-NN | 96.7 | 78.18 | 95.9 | 83.95 | 96.1 | 81.0 |
| | RUSBoosted Trees | 93.3 | 75.68 | 95.0 | 78.05 | 93.3 | 76.09 |
| Decision Trees | Fine Tree | 100 | **86.0** | 100 | 81.82 | 100 | 72.73 |
| | Medium Tree | 88.1 | 52.73 | 90.4 | 59.05 | 87.1 | 63.82 |
| | Coarse Tree | 95.8 | 81.32 | 94.5 | 84.23 | 93.8 | 80.64 |

for detecting ROIs associated with each astrocyte in the $Ca^{2+}$ imaging videos. The quality of segmentation results is usually only compared to the hand-drawn ground truths (ROIs that encompassed the cell somata). However, we fed the obtained time-series traces to the classification procedure, given that another set of ground truths was available regarding the type of cell treatment–with IgGs from non-ALS (12 videos) or from ALS patients (21 videos) [23]. Our findings point out to the interesting link between the criteria governing the growth of candidate ROIs and subsequent classification accuracy, that is most likely related to the features of $Ca^{2+}$ signals comprising feature images. Therefore, we expect our method to bridge the gap between ROI-based and event-based approaches in the analysis of the results from standard fluorescence video microscopy.

Interestingly, we observed that achieving better scores in segmentation results (in comparison to ground truths) did not necessarily correspond to an increase in classification performance. It is well known that astrocytes display complex morphology due to the exquisitely irregular, dynamic, and ultrathin (30–50 nm up to 200 nm) nature of distal compartments, broadly named processes or filopodia, comprising the majority of the cell volume (up to 80–85%) [29]. This effect is well described in brain slices and *in vivo*, respectively [26], however it might also be observed in cell cultures [27]. By being of nanoscale size and rather thin, the filopodia dimensions are below the diffraction limit of light, and thus cannot be resolved by light microscopy [29, 30]. Yet, a spatial buildup of $Ca^{2+}$ fluctuations is reflected in the occurrence of intra- and intercellular $Ca^{2+}$ waves, and their time-scale spans from hundreds of milliseconds to tens of seconds [30, 31]. Filopodia structural changes surely influence calcium dynamics [29], and given that *in vitro* astrocytes generally have closer contacts with each other in comparison to *in situ* conditions, astrocytes in culture routinely exhibit intra- and intercellular calcium waves [32]. In fact, such waves are very suitable as a means of determining cell

boundaries by utilizing the multi-level thresholding method allowing for the progressive enlargement of the area of candidate cell regions.

We herewith relied on the bulk-loading of astrocytes with the $Ca^{2+}$-sensitive membrane-permeable dyes, such as Fluo-4, that lead to the investigation of their $Ca^{2+}$ transients obtained from video traces. Fluo-4 reveals even the small fluctuations in intracellular $Ca^{2+}$ with changes in fluorescence intensity throughout the entire cell [33], however, it is now known that most of the fluctuations occur within the fine processes [26–28]. In addition, due to the above-mentioned diffraction limit of light microscopy, newer photonics tools such as super-resolution microscopy, as well as the computational modeling, remain the only methodologies to target specific cellular compartments. This means not only resolving particular compartment geometry but also the molecular distribution and diffusion, which are deemed essential in such small volumes [29–31]. Nevertheless, for the bench to bedside application of an automatic screening system, it is imperative to use simpler technical solutions that are easily portable and do not require special training.

The measurement of $Ca^{2+}$ fluctuations from astrocytic soma is relatively simple, owing to the easy identification of the broad somatic area [28]. In addition, by carefully selecting ROIs, one could also acquire information on $Ca^{2+}$ signaling by imaging the fluorescence from the periphery of a cell [28]. However, constraining calcium signals in fixed spatial boundaries may result in signal detection inaccuracy or partial detection as it may become larger than or get out of the ROI possibly overlapping with ROIs of adjacent cells [28, 30]. To solve this issue, event-based algorithms have been developed by adopting the logic that an event presents an increase in fluorescent intensity that can be captured based on its dynamic changes in space and time such as related to spatial size, shape, propagation direction, duration, frequency, and amplitude [28, 29]. Thus, our method effectively bridges the gap between the ROI-based and event-based approaches, since after applying the segmentation stage for detecting boundaries of astrocytes and extracting traces, the feature images were created using the ROIs and the corresponding traces.

Several software solutions exist [27–30] that aim to improve the analysis of astrocyte microdomain calcium. Notably, CaSCaDe ($Ca^{2+}$ Signal Classification and Decoding) [34] uses machine-learning to identify calcium events. It is named so because each analysis step is dependent on the outcome of the previous step [34]. Briefly, similar to the method we proposed here, time-series images were first processed to remove background noise on the sum-intensity projected image. In the next step, such images were binarized using a threshold level of two standard deviations from noise. After smoothing with a Gaussian filter, regional maxima were determined as sites of putative microdomains, and such a binarized mask was further refined based on the additional criteria of the intensity level over time, duration, and size. Further on, the SVM algorithm was used to detect active microdomains after conducting a training on 75 parameters of each event extracted from one raw and two smoothed intensity profiles with different degrees of smoothness, while a set of 2500 fluorescence signals were manually categorized as positive or negative [34]. Although our method has a few parameter constraints that are optimizable, it takes the advantage of combining segmentation and classification-based approach for automated analysis of biomedical signals. For the multi-level thresholding phase it does not have to rely on manually selected ROIs and for the classification phase it does not have to rely on hand-crafted training parameters of each event. It demonstrates that those two approaches can be used complementarily in order to dynamically determine the best ROI size.

In conclusion, we have herewith demonstrated that our method can be used on the entire fluorescence imaging videos, depicting: 1) different experimental groups in terms of treatment, and 2) sequence-to-sequence differences in terms of experimental phases (baseline, acute

treatment, washout, and viability check). It thus provides a semi-autonomous tool for assessing segmentation parameters which allow for the best classification accuracy.

## Author Contributions

**Conceptualization:** Dunja Bijelić, Lidija Radenović, Abdulkerim Çapar, Bilal Ersen Kerman, Pavle R. Andjus, Andrej Korenić, Ufuk Özkaya.

**Data curation:** Gizem Dursun, Dunja Bijelić.

**Formal analysis:** Gizem Dursun.

**Funding acquisition:** Pavle R. Andjus.

**Investigation:** Gizem Dursun, Dunja Bijelić, Neşe Ayşit, Burcu Kurt Vatandaşlar.

**Methodology:** Gizem Dursun.

**Project administration:** Lidija Radenović, Pavle R. Andjus.

**Resources:** Ufuk Özkaya.

**Software:** Gizem Dursun, Abdulkerim Çapar.

**Supervision:** Abdulkerim Çapar, Ufuk Özkaya.

**Validation:** Abdulkerim Çapar, Ufuk Özkaya.

**Visualization:** Gizem Dursun.

**Writing – original draft:** Gizem Dursun, Neşe Ayşit, Burcu Kurt Vatandaşlar, Bilal Ersen Kerman, Andrej Korenić, Ufuk Özkaya.

**Writing – review & editing:** Gizem Dursun, Dunja Bijelić, Lidija Radenović, Abdulkerim Çapar, Bilal Ersen Kerman, Pavle R. Andjus, Andrej Korenić, Ufuk Özkaya.

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
