## [Decision Letter · Decision Letter 0]

22 Sep 2022

PONE-D-22-19330Combined segmentation and classification-based approach to automated analysis of biomedical signals obtained from calcium imagingPLOS ONE

Dear Dr. Korenić,

Thank you for submitting your manuscript to PLOS ONE. After careful consideration, we feel that it has merit but does not fully meet PLOS ONE’s publication criteria as it currently stands. Therefore, we invite you to submit a revised version of the manuscript that addresses the points raised during the review process.

We look forward to receiving your revised manuscript.

Kind regards,

Yiming Tang, Ph.D.

Academic Editor

PLOS ONE

Journal Requirements:

3. Thank you for including your ethics statement:  " Animal procedures were carried out in accordance with the strict protocols of the Ethics Committee for the Use of Laboratory Animals of the Faculty of Biology, University of Belgrade, Serbia (rsr. lic. 323-07-10457/2019-05) and in the compliance with the National ethics committee – SLASA, as well as the EU Directive (2010/63/EU) on the protection of animals used for scientific purposes. For the purpose of the AUTOIGG project (EC H2020 MSCA-RISE project No 778405) approval for the human subjects research was obtained (850/6), as well as participant consent form which was implemented in the study.".  

To comply with PLOS ONE submissions requirements, please provide the following information in the Methods section of the manuscript and in the “Ethics Statement” field of the submission form (via “Edit Submission”): 

*  Please indicate whether an animal research ethics committee prospectively approved this research or granted a formal waiver of ethics approval.*  Please enter the name of your Institutional Animal Care and Use Committee (IACUC) or other relevant ethics board. Also include an approval number if one was obtained.

*   If anesthesia, euthanasia, or any kind of animal sacrifice is part of the study, please include briefly in your statement which substances and/or methods were applied.

For additional information about PLOS ONE submissions requirements for ethics oversight of animal work, please refer to http://journals.plos.org/plosone/s/submission-guidelines#loc-animal-research 

Reviewers' comments:

Reviewer's Responses to Questions

**Comments to the Author**

1. Is the manuscript technically sound, and do the data support the conclusions?

Reviewer #1: Partly

Reviewer #2: Partly

2. Has the statistical analysis been performed appropriately and rigorously? 

Reviewer #1: No

Reviewer #2: Yes

3. Have the authors made all data underlying the findings in their manuscript fully available?

Reviewer #1: No

Reviewer #2: Yes

4. Is the manuscript presented in an intelligible fashion and written in standard English?

Reviewer #1: Yes

Reviewer #2: Yes

5. Review Comments to the Author

Reviewer #1: 1) “a semi-autonomous tool for the selection of the most optimal size of a ROI” No evaluation allows to know if the size of ROI is optimal. Formally, there are no optimization criteria.

2) In the abstract and in the summary, we do not know what are the classes used in the classification process.

3) Line 36: “The mean value of the F-score for all the data was above 0.80”. The value of the F-score is given before having defined the true positives, false positive and false negatives?

4) Line 160: “obtained binary mask was compared to the ground truths” it can be assumed that the ground truth associated with the segmented image is the original image. But how the comparison is made ?

5) How can you distinguish between the two scenarios for enlarging the area of candidate cell regions, the situation where single region is expanding and the situation where two cells are merging? A single region could be split into two (or more) if the intensity is not homogeneous in that region, how to know if it is the same region or two (or more) different regions.

6) Line 247 : “Shuffling and image resizing were implemented to increase the number of data that were used …” How shuffling and resizing increase the number of data ?

7) Line 251: “a shuffled feature image was downsized to 128×32”. We know that a feature image has as many rows as the number of detected ROIs and as many columns as time points. How did compare the number of rows and the number of columns in initial images to 128x32?

8) Line 255: “This data augmentation process is important both to make training invariant to the order of ROIs and to increase the number of images required to train a supervised classifier” We do not see how number of images is augmented ?

9) The number of patients and the number of healthy should be announced in the abstract and in the summary. Similarly the values of junction criteria, stopping and elimination criteria must be given in the method part and not in the results.

10) Figure 5: X axis title should be better specified (probably “IoM threshold value” instead of « Threshold value »).

11) In Fig. 5 caption: “The mean value of the F-score for all the data was about 0.86 throughout the first 5 threshold levels (Table 2).” But, the mean value of the F-score is not shown in figure 5 nor in table 2. In table 2, the F-value depends on the projection method. Please correct.

12) In table 2, what is the “threshold level” (IoM threshold?)

13) Line 305: “The mean values of the F-scores are also presented in Table 3 to compare the performances of different projection methods” This is not exact, table 3 shows “Comparison of F-score results for different junction criteria given IoM threshold level” Please correct.

14) The values of F-score in table 2 and table 3 are highly discordant. In table 3, for IoM = 0.5 (the value selected in this experiment, see line 298), the F-score is less than 0.4, whatever the value of junction criteria threshold. This is very different from the announced 0.8 value.

In the discussion, some assertions are not supported by the data despite what the authors assure:

15) Line 349: « Our findings point out to the interesting link between the particular size of the ROIs and subsequent classification accuracy that is most likely related to the astrocyte morphology and the related Ca2+ dynamics”. Size of Rois is not taken into account in the experiment since the ROI is characterized by the average value without taking into account the size. The dynamics are also not taken into account since this Ca2+ dynamics is lost by the temporal sub-sampling which is not designed to preserve the dynamics..

16) Line 360: « We have also observed in our experimental data sets that the amplitude and shape of the signal varied with the size and position of ROI”. There is no data to observe or confirm this.

17) The comparison of the proposed method with existing software that aim to improve the analysis of astrocyte microdomain calcium is insufficiently discussed. The essential difference is that the approach adopted by these software allow the refinement of the analysis by additional criteria, the intensity level over time, the duration, and the size. How can the proposed method which is based on the intensity alone and on the average by ROI be used in conjunction with these methods.

Reviewer #2: 1. How did you tune the hyperparameters in various machine learning classifiers?

2.The authors used multi-level thresholding for segmentation, should use some state of art method.

3. All the image quality is poor resolution, need to be improved.

6. PLOS authors have the option to publish the peer review history of their article (what does this mean?). If published, this will include your full peer review and any attached files.

Reviewer #1: **Yes: **Abdel-Kader Boulanouar

Reviewer #2: **Yes: **sneha

---

## [Author Response · Author response to Decision Letter 0]

3 Dec 2022

Reviewer #1: 

1) “a semi-autonomous tool for the selection of the most optimal size of a ROI” No evaluation allows to know if the size of ROI is optimal. Formally, there are no optimization criteria.

Thanks to the reviewer for drawing our attention to this vagueness in our manuscript. If in our study there were a defined degree of how optimal any given ROI is, it would be related to classification accuracy. This follows from the fact our fellow biology researchers were faced with the question of how much of the astrocyte's surface they should encircle when defining ROI (also related to the comment #4). We already collaboratively dealt with this topic in more detail in the Discussion. However, now we made changes to the lines 45–47 and 471–473, as well as added text 215–217.

2) In the abstract and in the summary, we do not know what are the classes used in the classification process.

We added required information – see line 37, 398. 

3) Line 36: “The mean value of the F-score for all the data was above 0.80”. The value of the F-score is given before having defined the true positives, false positive and false negatives?

We believe there is no need to define TP, FP and FN in the Abstract section. However, after giving this comment some thought, we decided to move these definitions from the Results section to the Methods section – see lines 245–253.

4) Line 160: “obtained binary mask was compared to the ground truths” it can be assumed that the ground truth associated with the segmented image is the original image. But how the comparison is made?

In our study, ground truths were ROIs marked by the experienced researcher with expertise in fluorescence microscopy (particularly calcium imaging) and astrocytes cell culture. First and foremost, we aimed to obtain segmentation results that are as close as possible to what the researcher had marked. However, during the testing of various segmentation methods, we observed an effect of the ROI size on the classification accuracy. Interestingly enough, the researcher confirmed to us that she and her predecessor had noted similar findings in calcium dynamics. However, these observations remained unpublished. She also brought up and highlighted a valid and plausible biological explanation involving astrocytes’ morphology, which we addressed in the Discussion.

Therefore, we made adjustments to the text – see lines 156–158, 169, 215–217.

5) How can you distinguish between the two scenarios for enlarging the area of candidate cell regions, the situation where single region is expanding and the situation where two cells are merging? A single region could be split into two (or more) if the intensity is not homogeneous in that region, how to know if it is the same region or two (or more) different regions.

Before the MLT step, we implemented mean filter for noise elimination and image smoothing. This filter ensures that a region's intensity variability is reduced, resulting in more homogeneous regions. Moreover, MLT starts from the maximum intensity, therefore, while being expanded, regions can only merge, not split. See lines 200, 207–209.

6) Line 247 : “Shuffling and image resizing were implemented to increase the number of data that were used …” How shuffling and resizing increase the number of data ?

Each feature image represents a single Ca2+ imaging video where, depending on the number of observable cells within the microscope’s field of view, the number of ROIs differs. Since during a single recording (experiment) astrocytes are treated with only one sample (non-ALS vs. ALS IgGs), all ROIs in the Ca2+ imaging video will belong to the same class. Therefore, shuffling was implemented to increase the number of data used for classification, as well as to make training invariant to the order of ROIs. After shuffling, the resulting image looks as if it was obtained from another experiment belonging to the same class. Subsequently, image resizing was used to standardize the size of all feature images for the classifiers (see the answer to the next comment).

Changes were made to the text in lines 267–280.

7) Line 251: “a shuffled feature image was downsized to 128×32”. We know that a feature image has as many rows as the number of detected ROIs and as many columns as time points. How did compare the number of rows and the number of columns in initial images to 128x32?

Rows and columns of the feature image are related to the number of ROIs (cells) in calcium videos and trace length, respectively. If the feature size is variable as in our study, then it is acceptable to select the mean length of the feature size as the normalized length. The normalized features are then fed to the machine learning algorithm. Therefore, the normalized height of the feature image was the average number of ROIs per video, which was around 32. The normalized width of the feature image was selected to be 128 to decrease the feature size with regard to the small number of samples in our dataset. It is known that more data samples are needed to train big sized feature vectors. We have selected this number to downsize the traces without losing distinct features, such as extremum points, of the traces.

Changes were made to the text in lines 267–280.

8) Line 255: “This data augmentation process is important both to make training invariant to the order of ROIs and to increase the number of images required to train a supervised classifier” We do not see how number of images is augmented?

A distinction should be made between the images that are actual frames in the video, and the feature image, which is comprised of the mean values of pixel intensities coming from ROIs in the video. Consequently, ‘feature’ refers to the various features of the calcium traces belonging to a single class (to be learned by the classifier). ‘Image’ refers to the pixel intensities.

9) The number of patients and the number of healthy should be announced in the abstract and in the summary. Similarly the values of junction criteria, stopping and elimination criteria must be given in the method part and not in the results.

We made the necessary corrections regarding number of videos in each group (see lines 29–30, 146–147, 305–306, 398). However, the values of junction criteria, stopping and elimination criteria were obtained as a result of optimization process. We had written that they were pre-determined, but in a sense that segmentation step comes before the classification step. Hence, we made the necessary modifications after we realized that our usage of such term was imprecise (see lines 213, 325–328, 344–345, 360–361).

10) Figure 5: X axis title should be better specified (probably “IoM threshold value” instead of « Threshold value »).

We made the suggested corrections.

11) In Fig. 5 caption: “The mean value of the F-score for all the data was about 0.86 throughout the first 5 threshold levels (Table 2).” But, the mean value of the F-score is not shown in figure 5 nor in table 2. In table 2, the F-value depends on the projection method. Please correct.

We made the necessary corrections, please see line 340.

12) In Table 2, what is the “threshold level” (IoM threshold?)

We made the necessary corrections.

13) Line 305: “The mean values of the F-scores are also presented in Table 3 to compare the performances of different projection methods” This is not exact, table 3 shows “Comparison of F-score results for different junction criteria given IoM threshold level” Please correct.

We made necessary corrections.

14) The values of F-score in table 2 and table 3 are highly discordant. In table 3, for IoM = 0.5 (the value selected in this experiment, see line 298), the F-score is less than 0.4, whatever the value of junction criteria threshold. This is very different from the announced 0.8 value.

We made necessary corrections to both Table 2 and 3.

In the discussion, some assertions are not supported by the data despite what the authors assure:

15) Line 349: « Our findings point out to the interesting link between the particular size of the ROIs and subsequent classification accuracy that is most likely related to the astrocyte morphology and the related Ca2+ dynamics”. Size of ROIs is not taken into account in the experiment since the ROI is characterized by the average value without taking into account the size. The dynamics are also not taken into account since this Ca2+ dynamics is lost by the temporal sub-sampling which is not designed to preserve the dynamics.

We propose to soften this statement as we would only be able to link the criteria governing the growth of candidate ROIs to the accuracy of the subsequent classification. Also, we agree with the reviewer that referring to calcium dynamics is slightly overreaching statement. In essence, we can only claim that changes in classification accuracy are related to the changes in feature images (i.e., calcium traces resulting from ROIs). We do want to retain the discussion of potential biologically based reasons for this in the text, though.

Therefore, we made adjustments to the text – see lines 295–297, 399–402, 471–473.

16) Line 360: « We have also observed in our experimental data sets that the amplitude and shape of the signal varied with the size and position of ROI”. There is no data to observe or confirm this.

We apologize for not clearly stating that these are unpublished observations. It was these findings that prompted us to make a connection between unsupervised segmentation and classification accuracy. Nevertheless, the sentence has been removed since unpublished results are not preferable.

17) The comparison of the proposed method with existing software that aim to improve the analysis of astrocyte microdomain calcium is insufficiently discussed. The essential difference is that the approach adopted by these software allow the refinement of the analysis by additional criteria, the intensity level over time, the duration, and the size. How can the proposed method which is based on the intensity alone and on the average by ROI be used in conjunction with these methods.

The method suggested in this study is not meant to be the most cutting-edge method in competition with existing software. Those were designed and tuned by relying either on more raw data or on handcrafted parameters. Also, it was not intended to be used for microdomain analysis. Our method was devised to allow automated video segmentation in an unsupervised fashion, as well as to indicate the relationship between the segmentation parameters and classification accuracy. Given that the whole length of the trace (although downsized) was used during the classification step, all signal parameters are taken into account as features (not only signal intensity).

Reviewer #2: 

1. How did you tune the hyperparameters in various machine learning classifiers?

We thank the reviewer for this comment, especially because we noticed an information missing about the software we used. Various machine learning classifiers were trained with The Classification Learner app in MATLAB ver. 2022a (The MathWorks, Inc., Natick, Massachusetts, United States), which does the job of tuning hyperparameters. We added this information to the lines 141-142.

2.The authors used multi-level thresholding for segmentation, should use some state of art method.

As a matter of fact, we used other segmentation methods, namely histogram equalization and active contour model, as well as their combination. Their performance, however, did not turn out to be as good as it was with MLT. Because we felt that, for instance, the description of criteria optimization was more important to elaborate on, we did not want to clog the text with inadequate methods. Nevertheless, perhaps one possibility would be to provide a new table as supplementary material to show how MTL and other segmentation techniques compare to one another. 

It is also worth mentioning that there are powerful deep learning-based image segmentation methods available recently. But, since they require a significant amount of training data, they are not suitable for our study. Working with a limited number of images would cause underlearning of the neural model. Also, some of the available methods had been developed for neurons that have a different morphology than astrocytes, not to mention calcium dynamics.

3. All the image quality is poor resolution, need to be improved.

The uploaded images have all been double-checked again for quality and resolution, and they are all in accordance with the journal requirements.

---

## [Decision Letter · Decision Letter 1]

19 Jan 2023

Combined segmentation and classification-based approach to automated analysis of biomedical signals obtained from calcium imaging

PONE-D-22-19330R1

Dear Dr. Korenić,

We’re pleased to inform you that your manuscript has been judged scientifically suitable for publication and will be formally accepted for publication once it meets all outstanding technical requirements.

Kind regards,

Yiming Tang, Ph.D.

Academic Editor

PLOS ONE

Additional Editor Comments (optional):

Reviewers' comments:

Reviewer's Responses to Questions

**Comments to the Author**

1. If the authors have adequately addressed your comments raised in a previous round of review and you feel that this manuscript is now acceptable for publication, you may indicate that here to bypass the “Comments to the Author” section, enter your conflict of interest statement in the “Confidential to Editor” section, and submit your "Accept" recommendation.

Reviewer #1: All comments have been addressed

Reviewer #2: All comments have been addressed

2. Is the manuscript technically sound, and do the data support the conclusions?

Reviewer #1: (No Response)

Reviewer #2: Yes

3. Has the statistical analysis been performed appropriately and rigorously? 

Reviewer #1: (No Response)

Reviewer #2: Yes

4. Have the authors made all data underlying the findings in their manuscript fully available?

Reviewer #1: (No Response)

Reviewer #2: Yes

5. Is the manuscript presented in an intelligible fashion and written in standard English?

Reviewer #1: (No Response)

Reviewer #2: Yes

6. Review Comments to the Author

Reviewer #1: (No Response)

Reviewer #2: The authors addressed the review comments and hence the manuscript can be accepted in its current form.

7. PLOS authors have the option to publish the peer review history of their article (what does this mean?). If published, this will include your full peer review and any attached files.

Reviewer #1: **Yes: **Abdel-Kader Boulanouar

Reviewer #2: No

---

## [Editor Report · Acceptance letter]

27 Jan 2023

PONE-D-22-19330R1 

Combined segmentation and classification-based approach to automated analysis of biomedical signals obtained from calcium imaging 

Dear Dr. Korenić:

I'm pleased to inform you that your manuscript has been deemed suitable for publication in PLOS ONE. Congratulations! Your manuscript is now with our production department. 

Kind regards, 

on behalf of

Professor Yiming Tang 

Academic Editor

PLOS ONE